CUTTING EDGE

# Using mobile sequencers in an academic classroom

**Abstract** The advent of mobile DNA sequencers has made it possible to generate DNA sequencing data outside of laboratories and genome centers. Here, we report our experience of using the MinION, a mobile sequencer, in a 13-week academic course for undergraduate and graduate students. The course consisted of theoretical sessions that presented fundamental topics in genomics and several applied hackathon sessions. In these hackathons, the students used MinION sequencers to generate and analyze their own data and gain hands-on experience in the topics discussed in the theoretical classes. The manuscript describes the structure of our class, the educational material, and the lessons we learned in the process. We hope that the knowledge and material presented here will provide the community with useful tools to help educate future generations of genome scientists.

**SOPHIE ZAAIJER, COLUMBIA UNIVERSITY UBIQUITOUS GENOMICS 2015 CLASS AND YANIV ERLICH***

## Introduction

The last decade has witnessed dramatic changes in the field of genomics with the advent of high-throughput DNA sequencing technologies. Sequencers have become the ultimate tool for a wide range of applications, from prenatal genetic screens and microbe identification to forensic sciences and autopsies. As such, genomics requires interdisciplinary thinking that involves concepts from molecular biology, statistics, computer science, and ethical and societal issues. Previous work has highlighted the benefit of hands-on training to help students put these concepts into context (*Altman 1998*; *Donovan, 2008*; *Reisdorph et al., 2013*; *Magana et al., 2014*). Hands-on training is also the preferred learning style of the Millennial generation, which currently makes up the majority of undergraduate and graduate students. Research has shown that people in this generation are technology focused, work most effectively in groups, and absorb information most efficiently by kinesthetic learning (learning by doing; *Shapiro et al., 2013*; *Evans et al., 2015*; *Linderman et al., 2015*).

Here, we describe our experience of using mobile DNA sequencers in the classroom to facilitate hands-on learning. Our class focused on the newest sequencing technology: the MinION by Oxford Nanopore Technologies (ONT). Unlike other sequencing technologies that are static and require a laboratory setting, the MinION sequencer is slightly larger than a typical USB stick and only requires a laptop to run (*Figure 1A and B*; *Gardy et al., 2015*; *McIntyre et al., 2015*; *Erlich, 2015*).

## Overview of the Ubiquitous Genomics class

We developed a course for Columbia University entitled 'Ubiquitous Genomics' that brings portable sequencing to the classroom. The Computer Science department offered the course as an elective. Of the 20 students who enrolled in the course, 50% were studying towards a bachelor's, 30% towards a master's degree, and 20% were enrolled in a PhD program. The majority (~60%) of students were enrolled in a computer science program, and the rest were enrolled in other programs, including electrical engineering, environmental health science and biomedical informatics. The class had no prerequisites, but nearly all students had some programming experience and about a third of the students had taken at least one class in computational

*For correspondence: yaniv@cs.columbia.edu

Group author details: Columbia University Ubiquitous Genomics 2015 class See page 8

biology. Students with computational biology experience performed slightly better in our class.

The course consisted of 13 meetings (one two-hour class per week) and was separated into a theoretical section and an applied section (*Supplementary file 1*). The theoretical section overviewed sequencing technologies and their potential uses in medicine, bio-surveillance, forensics, and ethical aspects of DNA sequencing, such as genetic privacy and the ability of participants to comprehend risks and potential harm. The aim of the theoretical section was to create a common ground for the group of students with diverse majors and background knowledge. The format was an interactive seminar where the class discussed one or two recent research papers.

The applied section comprised two three-week blocks of "hackathons" that included MinION sequencing, data analysis and an assignment. We estimate the consumable costs of a hackathon to be on the order of $1,000 per team per assignment (*Table 1*). However, nearly 90% of the cost is due to the MinION sequencer and any reduction in its price will affect the projection of the costs. We decided to use the term hackathon to convey to the students that, unlike a regular course lab, the questions were open-ended and even we (the instructors) did not always know the answers or the best tools to solve the assignments. In the first hackathon, entitled "from snack to sequence", the students received unlabeled DNA collected from food and supermarket ingredients. They had to use the sequencers to collect the DNA data and devise a pipeline to infer the ingredients. In the second hackathon, called "CSI Columbia", the students sequenced several human DNA samples without knowing the identity of the samples. The hackathon focused on collecting data from these samples and students were encouraged to try any possible method they could imagine to generate investigative leads.

## The hackathon structure

To address our teaching goals, we set the three week hackathon cycle as follows: in the first week of each hackathon-block, the students met for a ~3-hour session, in which they worked in groups to set up the MinION sequencer, generate data, and start strategizing about the best approach to answer the assignment. In the second week, we had a meeting with the students to discuss technical issues related to the assignment, such as the best approach to identify an organism from MinION data. Each group had to explore a different approach and to report the results in a 5-minute presentation to the rest of the class. In the final class of each hackathon-block, the students presented their results and turned in their written assignments (*Supplementary file 1*).

Naturally, the most challenging classes to prepare for were the MinION sessions. We employed several strategies to maximize the hands-on experience of the students within the time constraints of the class (*Figure 2*):

A week before the hackathon, the students were instructed to form groups of 4–5 people. We encouraged them to form groups with diverse skill sets (e.g. combinations of biology backgrounds and computer science backgrounds).

Several days before the hackathons, the instructors prepared the DNA libraries for the class. We decided to do this part ourselves and not as part of the training, since genomic DNA extraction and ONT library preparation takes ~4 hr (*Supplementary file 2*). It was not realistic to include these steps as part of the hackathon given the time limits (although this might change with the advent of the automated library preparation device, the VolTRAX).

Each hackathon started by tuning student expectations; we reminded the students about the experimental nature of the events. We communicated clearly that they should anticipate technical issues and that we would be surprised if everything went smoothly. This helped to reduce frustration for the students, who are accustomed to interacting with mature technology in day-to-day life. We continued with a 45-minute lecture about the goals of the hackathon and background material, such as how the DNA libraries were prepared, the MinION software interface features, and the base-calling pipeline (see *Supplementary files 3–6* for assignments and PowerPoint slides).

Next, we had students practice pipetting. The loading of reagents onto the MinION flow cell requires good pipetting skills; otherwise, the yield may be substantially lower. As most of our students had never touched a pipette before, we allowed them to practice loading water onto used MinION flow cells until they were comfortable pipetting with precision.

Armed with a protocol, the students were fully responsible for generating the data with minimal assistance. They connected the devices

**A**

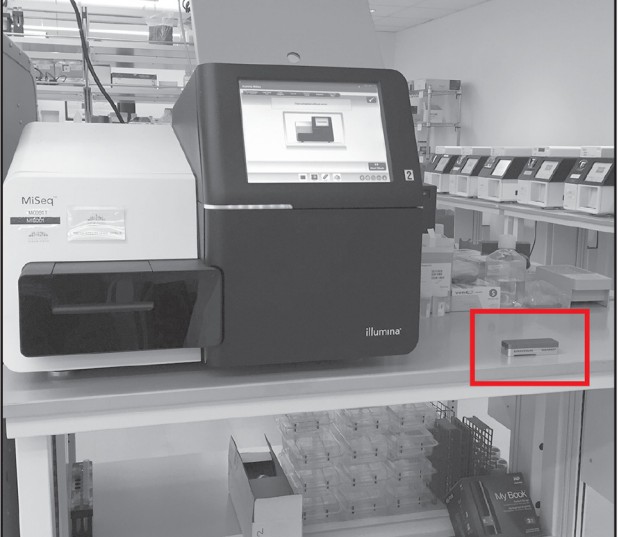

**B**

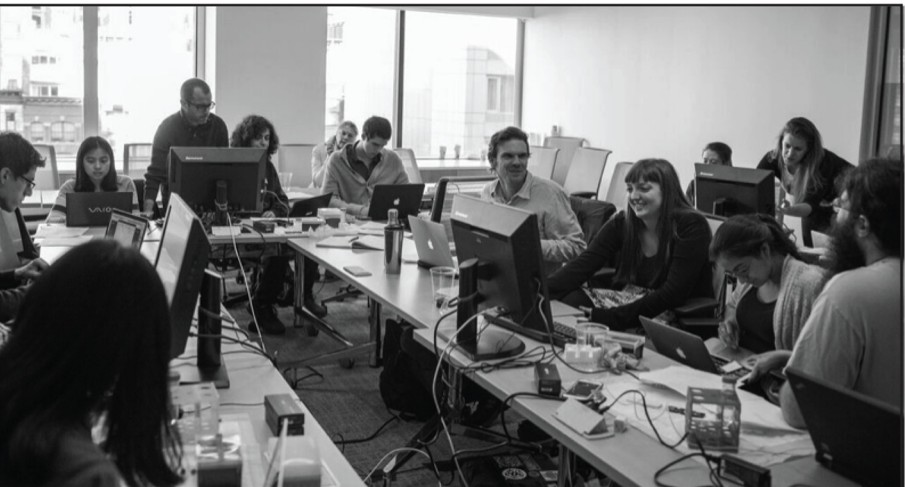

**Figure 1.** The Ubiquitous Genomics class. (**A**) Illumina MiSeq benchtop sequencer (left) and the MinION sequencer made by Oxford Nanopore Technologies (right; red rectangle). (**B**) The class during a hackathon.

to the computers, activated the relevant programs, loaded the priming mix (dubbed 'fuel') and DNA libraries onto the flow cells, and launched the sequencing run using MinKnow. Once data was generated, they monitored the progress of the sequencing run. After checking quality measures, the sequencers were left unattended for 48 hours to generate data according to the ONT protocol.

After data generation, we instructed the students to complete an assignment, which was divided into two milestones (***Supplementary files 5*** and ***6***). The first

milestone was to report on the technical performance of the MinION sequencer, such as the total reads, the read length distribution, DNA library quality, and the read quality scores over time. The aim of the quality control analysis was to guide the students on how to approach large genomic data sets. The second milestone focused on an actual scientific problem (see below). For each milestone, the students had to submit a written report and a GitHub link to their code (an example: https://github.com/dspeyer/ubiq_genome). Each hackathon concluded with a 10-minute talk by each group. All

**Table 1.** MinION consumables:Total cost estimate (in US Dollars) is for one MinION run per team. For the complete list of equipment and consumables required for organizing a hackathon, please see the following link: https://nanoporetech.com/uploads/community/Equipment_and_consumables_vC_with_FAQ_Sep2015.pdf

| Company | Product | Cat no | Price per unit (USD) | Unit quantity | Amount needed for ONT protocol | Cost |
|---|---|---|---|---|---|---|
| Covaris | g-TUBE | 520079 | $275 | 10 | 1 | $27.50 |
| NEB | Ultra™ End Repair/dA-Tailing Module | E7442S | $225.00 | 72 ul | 3 μl | $9.40 |
| Agencourt | AMPure XP | A63880 | $315.00 | 5 ml | 60 μl | $3.78 |
| Thermo Fisher | Dynabeads MyOne Streptavidin C1 | 65001 | $475.00 | 2 mL | 50 μl | $12 |
| NEB | Blunt/TA Ligase Master Mix | M0367S | $95 | 250ul | 50 μl | $19 |
| | Tubes/ pipette-tips/$H_2O$/ ethanol etc | | | | | ~$10 |
| ONT | Flow-cell Reagent kit | | $900 | 1 | 1 | $900 |
| Projected cost per team per run: | | | | | | $981.68 |

relevant teaching material is provided under the Creative Commons Attribution-Share Alike 4.0 International License.

## Hackathon project 1: Snack to sequence

The first hackathon was called "from snack to sequence". It was inspired by several food scandals, such as the horsemeat found in ready-made meals that were labeled as beef throughout Europe in 2013, as well as the revelation that a number of sushi restaurants in New York City claimed to be selling white tuna but in reality were serving escolar. Based on this issue, we wanted to introduce students to the identification of species in different food items.

We prepared five sequencing libraries from dishes purchased at local restaurants and raw food products that were purchased at a supermarket. The DNA libraries were a mix of multiple ingredients (like raw beef and tomato). We set out to address the following questions with the students: a) Can you identify the species in a food sample using MinION sequencing, without prior knowledge? b) Can you quantify the composition of the different ingredients? c) What is the minimal sequencing runtime required to detect the ingredients of the sample?

After generating the data in the hackathons, we devoted the next class to exploring a diverse number of sequencing algorithms that could be used for species identification. Importantly, Oxford Nanopore's 'What's In My Pot' species identification workflow does not support the identification of eukaryotic samples (*Juul et al., 2015*) and the students had to find alternatives.

The consensus among the students of the class was that a Basic Local Alignment Search Tool (BLAST) was the best option.

Most groups were able to identify the species within the dish. One interesting discussion resulted from the two groups that sequenced samples putatively containing beef. The top BLAST hit was for bighorn sheep (*Ovis canadensis*), whereas the domesticated sheep (*Ovis aries*) or cow (*Bos taurus*) was returned with lower alignment quality values. The identification of bighorn sheep was suspicious, since this animal is not domesticated. Cow is part of the *Bovidae* family, as are the bighorn and domesticated sheep. The students reasoned that the sample could be from a family member and selected the domesticated sheep as the most likely candidate. A surprising finding was the detection of DNA from the parasites *Babesia bigemina*, *Wuchereria bancrofti* and *Onchocerca ochengi* in the raw beef samples (at least two or more reads per parasite). These findings led to a vivid discussion in the class on food safety. (Note: After reading a previous version of this manuscript on bioRxiv, Steven Salzberg noted that the Genbank sequences of these parasites are likely to be contaminated with cow DNA. Thus, the BLAST matches to these parasites do not conclusively indicate that they were present in the food samples.)

Overall, this hackathon was academically apt for the level of the students. The only technical challenge the students repeatedly encountered was how to BLAST a large number of query sequences using the application programming interface (API). They had to find creative solutions, such as mirroring the National Institutes of

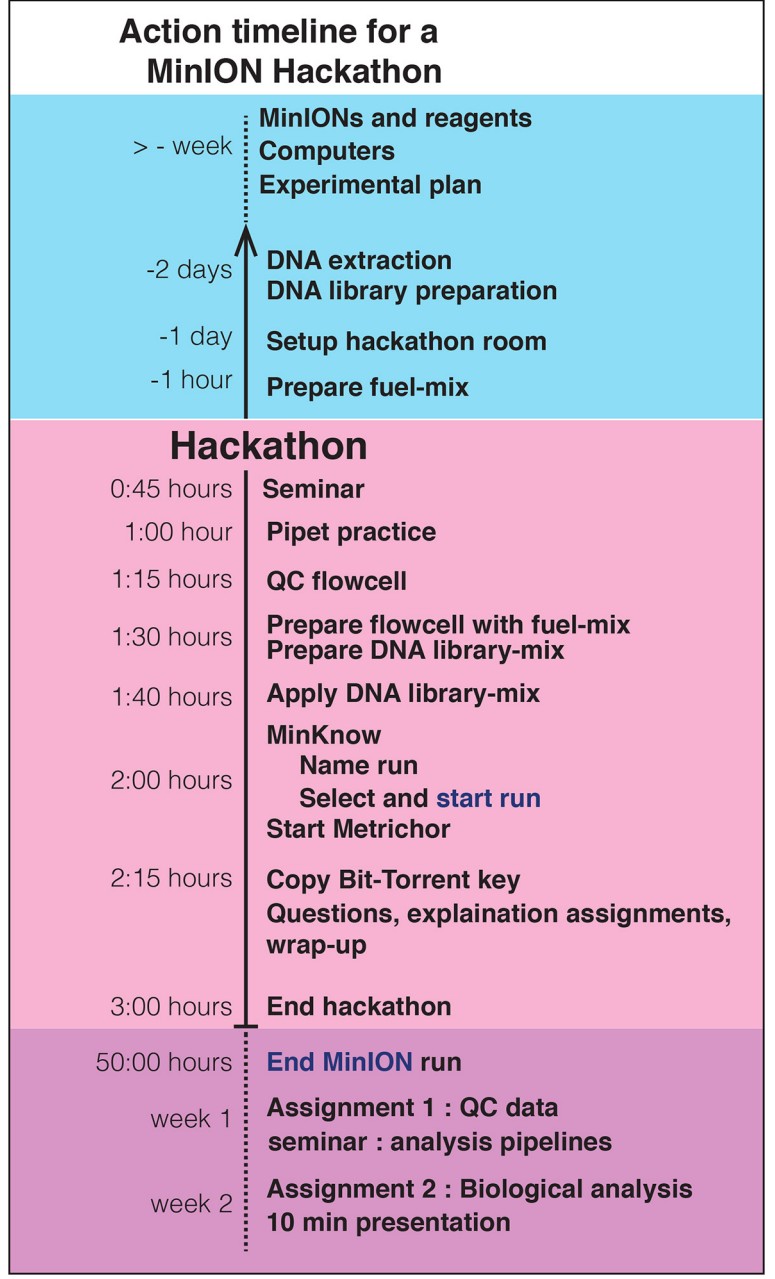

**Figure 2.** A detailed workflow for running a hackathon using a MinION sequencer.

Health (NIH) BLAST to a private server and tweaking the input parameters to make it possible to search a large number of long MinION reads.

## Hackathon project 2: CSI Columbia
For the second hackathon, we explored the identification of individuals using ultra low coverage genome sequencing with the MinION. In forensics, DNA evidence identification relies on the analysis of the 13 well-characterized Combined DNA Index System (CODIS) short tandem repeat (STR) loci (*Kayser and de Knijff 2011*). However, theoretical analysis has suggested that a small number (30–80) of common single nucleotide polymorphisms that are inherited independently of each other are sufficient for positive identification (*Lin et al., 2004*). The aim of this hackathon was to test whether it would possible

to use this technique to identify individuals using MinION shotgun sequencing with extremely shallow coverage. We also encouraged the students to test various methods to identify the person, such as examining the mitochondrial haplogroup, the sex of the person, and estimating his or her ancestry. In any case, our expectations were focused on their scientific decision process rather than the answer, and the students were encouraged to send the instructors questions when they required help.

Two groups sequenced a DNA library prepared from genomic DNA from Craig Venter (Levy et al., 2007), one group sequenced a Hap-Map sample from the 1000 Genomes Project, and two groups sequenced the genomic DNA of one of the authors (YE). We chose these individuals because of their publically available DNA reference data. The students initially did not know the identity of the sequenced genome, but in a later stage of the hackathon we told them that their sample was either one of the following individuals: Craig Venter, Jim Watson, the author (YE), or a participant of the 1000 Genomes Project.

The students found this assignment much more challenging than the previous one. Of the five groups, one was able to correctly identify their input sample (Craig Venter). The students tried an impressive array of tools but their main challenge was data wrangling. They had to convert their data to various formats in order to test different tools just to realize that the tools did not perform as expected or were poorly documented, wasting a significant amount of time. Interestingly, some of the undergraduate students told us later that this was the first time they were exposed to an open-ended real-world research problem and that this task gave them a better understanding of academic research. The students also suggested that more discussions between the groups during the hackathon could have helped to solve some of the technical problems. This can be done using online communication tools (like Facebook or a Piazza website). Future instructors of this hackathon can circumvent some of the difficulties by restricting the scope of the analysis. For example, instead of instructing the students to generate any possible identity lead, students could be told to focus only on ancestry analysis from shotgun sequencing or sequence specific regions such as the mitochondria for a more structured analysis.

## Lessons learned from conducting MinION hackathons

**Prepare spare parts:** We experienced multiple technical difficulties in the 10 intended MinION runs (five groups over in two hackathons). Three flow cells had an insufficient pore number (<51) and had to be replaced. In another event, a computer failed to connect with any MinION instruments despite a working USB 3.0 port. During the hackathon, there is little time to troubleshoot. It is therefore crucial to anticipate scenarios of failure and have spare parts (i.e. computers, flow cells, fuel mix, and DNA library).

**Consider back-up data:** As part of testing our hackathon setting, we sequenced some of the DNA libraries with the MinION before the actual event. The data generated from these tests was kept to have a contingency plan in case none of the MinIONs worked at the time of the hackathon. This way students would still have data to analyze, and the course progression would not be jeopardized. While we fortunately did not have to use this data, we encourage MinION hackathon organizers to consider this option.

**Expect variability in the amount of data:** The yield of the MinION sequencers varied between runs. The experimental design and the questions posed during each hackathon should be compatible with both a low and a high sequencing yield.

**Locate appropriate computers:** One of our main challenges was to procure five computers that matched ONT specifications. Our department is almost entirely Mac-based, whereas the current ONT specification requires a Microsoft Windows computer. We tried installing Windows virtual machines on our Macintosh computers but found this solution unreliable, presumably due to the fast data transmission rates of the sequencers. The students' computers also fell short of the specifications required by ONT, such as having a solid-state drive. MinION hackathon organizers should keep in mind that locating multiple appropriate computers can be a time-demanding task.

**Network:** ONT sequencing requires an Internet connection for base-calling. We connected the five computers to a regular network hub using a standard Ethernet protocol. We did not experience any issues.

**Use free tools for data transfer:** MinION sequencing can result in large data folders. We looked for a free program to automatically transfer the data 48 hrs after the start of the run from

the sequencing laptop to the students' computers. Cloud-based products, such as Dropbox, do not support synchronizing this amount of data with their free accounts. As an alternative, we used the free version of BitTorrent Sync, which allows sharing of files over the P2P Bit-Torrrent network without a size limit. BitTorrent can be pre-installed on the workstation and can be synchronized with the student's personal computer by exchanging a folder-specific key. This solution for large files can be set up within a few minutes and prevents technical challenges.

## Questionnaire

We sought to learn more quantitatively about the views of students with respect to genomics and mobile sequencing. We asked them to answer a questionnaire before the first hackathon, when the students were exposed only to the theory of sequencing and its applications, and then three weeks later, after the completion of the first hackathon.

While our sample size is too small to draw statistical conclusions, we did learn from the trends in the answers. The hackathons seemed to have shaped a more realistic view of the technical challenges inherent to genomic applications. For instance, for the question "How long do you think it takes from sample preparation to sequencing results using MinION?", about 70% of the students answered 'one hour' (or less) before the hackathon; but after the hackathon, only 30% of the students thought it would take one hour. After the hackathons, students also thought that it would take more time for mobile sequencers to be used for health tracking by the general public and suggested lower costs for home sequencing applications. Despite discussing the ethical implications of DNA analysis quite extensively throughout the course, we did not observe changes in the students views on several ethical issues such as "Do you think it is ethical to sequence hair found on the street?" or "Do you think getting your genome sequenced is safe?" These trends suggest that the hackathon mainly shaped the students' technical understanding and demonstrated the value of hands-on experience as a way of helping them to develop realistic views of the challenges of new technologies.

## Concluding remarks

Mobile sequencing in the classroom proved to be a useful method for teaching students about the cross-disciplinary field of genomics and to contextualize genomic concepts. These devices are relatively inexpensive and do not require complicated equipment or designated lab space to be operated. As such, they dramatically reduce the barrier to classroom integration compared to other sequencing technologies.

The main focus of this manuscript was the use of mobile sequencing in the higher education system (undergraduate and postgraduate). Even though most students were computer science majors, it could be suitable for other majors such as molecular biology and pharmacy, and also for medical school students. We highly recommend instructors of students with limited programming backgrounds to design assignments that use existing data analysis pipelines (such as ONT's "What's in my pot" tool). It might also be useful to customize the assignments to the major of the students. For example, for biology students, the assignment could focus on taxonomy, while medical students could benefit from sequencing microbes that are known to cause disease. We also see the potential of using these devices in high school STEM curricula and enrichment programs. Such activities can expose pupils early in their training to the fascinating world of DNA and serve as an educational springboard to study other disciplines such as math, computer science and chemistry. We hope that the resources and experience outlined in this manuscript will help to facilitate the advent of these programs.

### Acknowledgements

We thank James Brayer, Michael Micorescu and Zoe McDougall from Oxford Nanopore Technologies for technical support and for providing the reagents to the class. We thank Steven Salzberg for useful comments and discussion on our bioRxiv pre-print, and John Kender and Ronit Zaks for exposing us to the concept of millennial specific education. We also thank Dina Zielinski for useful comments on the manuscript. YE holds a Career Award at the Scientific Interface from the Burroughs Wellcome Fund. This work was partially supported by a National Institute of Justice (NIJ) grant 2014-DN-BX-K089 (YE and SZ).

**Sophie Zaaijer** is in the Department of Computer Science, Fu Foundation School of Engineering, Columbia University, New York, United StatesNew York Genome Center, New York, United States

**Yaniv Erlich** is in the Department of Computer Science, Fu Foundation School of Engineering,

Columbia University, New York, United StatesNew York Genome Center, New York, United StatesDepartment of Systems Biology, Center for Computational Biology and Bioinformatics, Columbia University, New York, United States

## Additional information

### Group author details

Columbia University Ubiquitous Genomics 2015 class

Maya Anand; Anubha Bhargava; Anne Bozack; Michael Curry; Alexander Kalicki; Xinyi Li; Katie Lin; Michael Nguyen; Diego Paris; Cheyenne Parsley; Robert Piccone; Garrett Roberts; Daniel Speyer; David Streid; Brian Trippe; Shashwat Vajpeyi; Boyu Wang; Lilly Wang; Tia Zhao; Liyuan Zhu

**Author contributions:** SZ, YE, Conception and design, Acquisition of data, Analysis and interpretation of data, Drafting or revising the article

***Competing interests:*** The authors declare that no competing interests exist.

## Additional files

### Supplementary files

• Supplementary file 1. Class website information.
This teaching material is provided under the Creative Commons Attribution-Share Alike 4.0 International License (© copyright Zaaijer et al, 2016).

• Supplementary file 2. Protocol for setting up a hackathon using MinION.
This teaching material is provided under the Creative Commons Attribution-Share Alike 4.0 International License (© copyright Zaaijer et al, 2016).

• Supplementary file 3. Student handout for Hackathon 1: Snack to sequence.
This teaching material is provided under the Creative Commons Attribution-Share Alike 4.0 International License (© copyright Zaaijer et al, 2016).

• Supplementary file 4. Student handout for Hackathon 2: CSI Columbia.

This teaching material is provided under the Creative Commons Attribution-Share Alike 4.0 International License (© copyright Zaaijer et al, 2016).

• Supplementary file 5. Student handout for the final project.
This teaching material is provided under the Creative Commons Attribution-Share Alike 4.0 International License (© copyright Zaaijer et al, 2016).

• Supplementary file 6. Teaching slides for Hackathon 1: Snack to sequence.
This teaching material is provided under the Creative Commons Attribution-Share Alike 4.0 International License (© copyright Zaaijer et al, 2016).

• Supplementary file 7. Teaching slides for Hackathon 2: CSI Columbia.
This teaching material is provided under the Creative Commons Attribution-Share Alike 4.0 International License (© copyright Zaaijer et al, 2016).

### Funding

| Funder | Grant reference number | Author |
|---|---|---|
| National Institute of Justice | 2014-DN-BX-K089 | Sophie Zaaijer Yaniv Erlich |

The funders had no role in study design, data collection and interpretation, or the decision to submit the work for publication.

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
