## [Decision Letter]

Thank you for submitting your article "Integration of mobile sequencers in an academic classroom" to *eLife* for consideration as a Feature Article. Your article has been reviewed by Samuel Donovan, Michael Linderman, and Nicholas Loman, and the evaluation was overseen by a member of the *eLife* staff (Sarah Shailes, Assistant Features Editor).

The peer reviewers and Dr Shailes have drafted this decision letter to help you submit a revised version of the article.

Dr Shailes will also contact you separately about some editorial issues that you will need to address.

Summary:

This was an enjoyable and interesting report about a hands-on workshop to train students in evolution, genomics, sequencing and bioinformatics employing the MinION portable genome sequencer. For the course, students completed a six-week introductory block of seminar-style classes and then two three-week "hackathons" in which they performed experiments using the MiniION sequencer. The paper has the potential to raise awareness about, and provide a concrete example of, the types of authentic science that can be done in undergraduate classrooms. This is an area of intense interest in the science education community and many efforts are underway suggesting that there should be a broad audience interested in this report.

The report highlights several important design features that are worth sharing including the interdisciplinary nature of the course, the "hackathon" instructional framework, and the focus on data management skills. The reported exercises and "lessons learned" will be valuable to other educators considering running a similar course.

Before publication, the following comments should be addressed.

Essential revisions:

1) As it is currently organized, the manuscript focuses too narrowly on the details of the technology employed. The context of this course is important, but the broader potential for informing diverse educational settings is not as clear as it could be. Please, therefore, include a fuller discussion of the key features of the approach, for example:

1a) The audience. Was the course advertised as a "bioinformatics" course? Did it have any prerequisites? Did students without a bioinformatics background have any particular difficulties with the hackathon projects? One or two sentences about the framing of the course and the experiences of different kinds of students would help provide context and orient other educators considering similar techniques.

1b) Using a "hackathon" instructional model – share with the audience what the features of a hackathon are and how it structured the classroom.

1c) Tuning student expectations – this strikes me as essential when working on authentic tasks.

1d) At a certain level this curriculum could be criticized as a "canned lab" because of the ways that the samples were prepared, and the data collection is highly automated. The reviewers don't share this criticism but they think that it may be useful to articulate the learning goals in order to highlight all of the challenging work that was done even when the data generation did not involve a lot of student decisions.

As each of these points are introduced as challenges/opportunities you can follow-up with details from the course that demonstrate how those opportunities were leveraged for engagement and learning.

2) Please include more fine details of the methods used in the course to help other instructors to reproduce it themselves. It might be better to give an exhaustive "kit list" and perhaps even costs associated with setting up such a workshop.

3) Subsection “Questionnaire”: you report that "We did not observe changes before and after the hackathon for ethical issues…". How much discussion of potential ethical issues was included in the syllabus? That is, would any changes in such views (or not) result only from their personal experiences, or were students deliberately exposed to different viewpoints on those questions?

4) What are the ethical implications of using the instructors' own DNA samples in the CSI practical? A discussion on this point would be useful.

5) The CSI practical is much less likely to "succeed" than the food experiment because shallow coverage nanopore sequencing is quite error prone (10% error at best). There should be some discussion about making this practical more likely to succeed, either by doing something like mitochondrial sequencing or some bioinformatics method of imputation that is sensitive to nanopore error model.

---

## [Author Response]

*Summary:*

*This was an enjoyable and interesting report about a hands-on workshop to train students in evolution, genomics, sequencing and bioinformatics employing the MinION portable genome sequencer. For the course, students completed a six-week introductory block of seminar-style classes and then two three-week "hackathons" in which they performed experiments using the MiniION sequencer. The paper has the potential to raise awareness about, and provide a concrete example of, the types of authentic science that can be done in undergraduate classrooms. This is an area of intense interest in the science education community and many efforts are underway suggesting that there should be a broad audience interested in this report. The report highlights several important design features that are worth sharing including the interdisciplinary nature of the course, the "hackathon" instructional framework, and the focus on data management skills. The reported exercises and "lessons learned" will be valuable to other educators considering running a similar course.*

We thank the reviewers for sharing with us the excitement about integrating MinION sequencers in the classroom and for taking the time to carefully read and comment on our manuscript.

*Before publication, the following comments should be addressed. Essential revisions: 1) As it is currently organized, the manuscript focuses too narrowly on the details of the technology employed. The context of this course is important, but the broader potential for informing diverse educational settings is not as clear as it could be. Please, therefore, include a fuller discussion of the key features of the approach, for example: 1a) The audience. Was the course advertised as a "bioinformatics" course? Did it have any prerequisites? Did students without a bioinformatics background have any particular difficulties with the hackathon projects? One or two sentences about the framing of the course and the experiences of different kinds of students would help provide context and orient other educators considering similar techniques.*

We thank the reviewers for pointing this out. We added to the opening paragraph of the section “Overview of the Ubiquitous Genomics class” an in-depth description of the students’ background and the class. We also added to the conclusion section a short discussion on adaptation of the course to students from other disciplines.

*1b) Using a "hackathon" instructional model – share with the audience what the features of a hackathon are and how it structured the classroom.*

Great suggestion. The revised manuscript has a new section called “hackathon structure” that provides details on the hackathon sessions. The first paragraph highlights the differences from a regular frontal class.

*1c) Tuning student expectations – this strikes me as essential when working on authentic tasks.*

We thank the reviewers for this comment, we agree and therefore moved this from “lessons learned” to the hackathon progress.

*1d) At a certain level this curriculum could be criticized as a "canned lab" because of the ways that the samples were prepared, and the data collection is highly automated. The reviewers don't share this criticism but they think that it may be useful to articulate the learning goals in order to highlight all of the challenging work that was done even when the data generation did not involve a lot of student decisions.*

We respectfully disagree with description that the hackathons are canned labs. The assignments had multiple open-ended tasks and even we, the instructors, did not know what the best set of tools to approach the task were or what the results would be. For example, we did not envision finding traces of cattle parasites in raw meat or identifying beef as big sheep horn or the best strategy to identify a person from MinION data.

The revised text stresses this point. We also included the learning goals in Supplemental Note 1.

*As each of these points are introduced as challenges/opportunities you can follow-up with details from the course that demonstrate how those opportunities were leveraged for engagement and learning. 2) Please include more fine details of the methods used in the course to help other instructors to reproduce it themselves. It might be better to give an exhaustive "kit list" and perhaps even costs associated with setting up such a workshop.*

We thank the reviewers for this excellent point. Table 1 provides a list of reagents and their cost per reaction.

*3) Subsection “Questionnaire”: you report that "We did not observe changes before and after the hackathon for ethical issues…". How much discussion of potential ethical issues was included in the syllabus? That is, would any changes in such views (or not) result only from their personal experiences, or were students deliberately exposed to different viewpoints on those questions?*

We apologize for not making this point more clear. We covered ethical issues in Class #4, including genetic privacy, the US GINA legislation, and potential harm to participants. In addition, ethical questions were raised naturally in other contexts such as human identification and the meaning of race versus ethnic group. We revised the text (second paragraph subsection “Overview of the Ubiquitous Genomics class”) to highlight that ethics was part of the curriculum.

*4) What are the ethical implications of using the instructors' own DNA samples in the CSI practical? A discussion on this point would be useful.*

The DNA that was used belongs to Dr. Yaniv Erlich, the leading instructor of the class. One of the main topics of Dr. Erlich’s research is genetic privacy. He volunteered his DNA with full consent and awareness of privacy issues. Therefore, we do not see any difference between using his genome to the other samples (e.g. Venter).

Beyond personal interest, we used Dr. Erlich’s DNA because his genome is publicly available on the Internet. We added a sentence that this was our inclusion criterion for DNA samples and also added a link where the genome can be found.

5) The CSI practical is much less likely to "succeed" than the food experiment because shallow coverage nanopore sequencing is quite error prone (10% error at best). There should be some discussion about making this practical more likely to succeed, either by doing something like mitochondrial sequencing or some bioinformatics method of imputation that is sensitive to nanopore error model.

Excellent point. We added several sentences to the closing paragraph of the CSI Columbia section that describe how to vary the scope of the task to make it less challenging.